Evaluation of circulating tumor cells as a prognostic biomarker for early recurrence in stage II–III breast cancer patients using CytoSorter® system: a retrospective study

Jin Lidan 1
http://orcid.org/0000-0001-7288-9425 Fan Wan-Hung 2
Luan Yi 3
Wu Meiqiong 2
Zhao Wenhe 1 whzhao@zju.edu.cn
1 Department of Surgical Oncology, Sir Run Run Shaw Hospital, Zhejiang University College of Medicine , Hangzhou , China
2 Hangzhou Watson Biotech , Hangzhou , China
3 Department of Clinical Laboratory, Sun Yat-Sen Memorial Hospital, Sun Yat-Sen University , Guangzhou , China
Coates Philip
Electronic publication date: 2021 Apr 29
Publication date: 2021
Volume: 9
Electronic Location ID: e11366
Received 2020 Dec 16; Accepted 2021 Apr 7
Copyright: © 2021 Jin et al.
Copyright year: 2021
Copyright holder: Jin et al.
License: This is an open access article distributed under the terms of the Creative Commons Attribution License, which permits unrestricted use, distribution, reproduction and adaptation in any medium and for any purpose provided that it is properly attributed. For attribution, the original author(s), title, publication source (PeerJ) and either DOI or URL of the article must be cited.
License URL: https://creativecommons.org/licenses/by/4.0/

Keywords: Breast cancer, Early recurrence, Circulating tumor cells, CytoSorter

Funding: The authors received no funding for this work.

==============================
Purpose

Circulating tumor cells (CTCs) are known to be associated with late recurrence and poor prognosis in breast cancer (BC). Different CTC enrichment platforms have different CTC cut-off values for poor prognosis. This study aimed to evaluate whether preoperative CTCs could be a prognostic factor for early recurrence of disease in BC patients with resectable tumors, and to ascertain the CTC cut-off value for early recurrence with CytoSorter® CTC system.

Methods

Thirty-six stage II and III BC patients who had preoperative (pre-op) CTC detection and underwent a mastectomy or lumpectomy for curative intent between January and May 2018 were enrolled in this retrospective study. CTC detection was performed using CytoSorter® CTC system. Correlations of patients’ demographics, clinicopathological characteristics, adjuvant therapies and CTCs with relapse and survival were evaluated.

Results

CTCs were detected in 32 out of 36 patients before surgery. Nine patients developed relapses during follow-up, and seven of them were distant recurrence. Univariate analysis showed that CTCs were correlated with two-year recurrence free survival (RFS) and distant RFS (D-RFS) (P = 0.013 and 0.029, respectively). Two-year RFS and D-RFS were 85.2% and 88.9%, respectively, for patients with <4 CTCs, while 44.4% and 55.6%, respectively, for patients with ≧4 CTCs. In multivariate analysis, only CTC was shown to be correlated with two-year RFS (HR: 0.219, 95% CI: [0.058–0.82], P = 0.024) and D-RFS (HR: 0.218, 95% CI [0.048–0.977], P = 0.047).

Conclusion

BC patients with pre-op CTCs ≥4 per four mL of blood have significantly reduced two-year RFS and D-RFS. A pre-op CTC cut-off of four per four mL of blood was found for CytoSorter® to identify BC patients with a higher risk for early recurrence.

Introduction

Breast cancer (BC) is the most frequent cancer among women both in China and worldwide, accounting for 17.1% of all cancer and 8.16% of cancer-related death in women in China in 2015 (Bray et al., 2018; Zheng et al., 2019). Primary treatment remedy for early BC is surgery and radiation, followed by systemic treatments, such as hormone therapy, chemotherapy, targeted therapy or combined multiple therapies (Goldhirsch et al., 2013). A precise treatment plan is determined by the molecular subtype of tumor and locoregional tumor load (Huang & Yin, 2018). As for patients with triple-negative (three receptors negative, including ER, PR and HER2) and HER2-positive (human epidermal growth factor receptor 2) early BC, a neoadjuvant therapy might be optional (Huang & Yin, 2018). Although it is believed that BC in early phase is potentially curable, still about 30% of patients develop disease recurrence (Ayala de la Peña et al., 2019). The timing of relapses fluctuates greatly, influenced by patients clinicopathological characteristics as well as received therapies (Takeuchi, Tsuji & Ueo, 2005). A relapse occurring within 5 years of initial treatment is commonly known as “early recurrence”, while appearing five or more years later is referred to as “late recurrence” (Takeuchi, Tsuji & Ueo, 2005). Late recurrence is more observed in patients with hormone receptor-positive breast cancer and is less common among patients with hormone receptor-negative diseases (Pan et al., 2017). A study with longitudinal 24 years of follow-up concerning the annual hazard rates (HR) of BC relapses has shown that the annualized HR of recurrence was highest within the first 5 years (10.4%), with a peak between years 1 and 2 (15.2%) in spite of hormone receptor status (Colleoni et al., 2016). However, beyond 5 years, patients with ER-positive (estrogen receptor) diseases had higher HR of recurrence and annualized HR remained elevated and stable, reaching a second peak between years 16 and 18 (Colleoni et al., 2016). A recent study showed how adjuvant chemotherapy would affect recurrence of BC. HR of recurrence for patients with surgery only exhibited two early sharp peaks at 9 and 33 months, a wide intermediate one spanning from about 50 to 90 months and a late peak at 115–120 months (Demicheli et al., 2020). When adjuvant chemotherapy was introduced, HR for early recurrence was reduced, leaving an early residual peak at about 18 months and two small intermediate peaks at about 50 and 80 months (Demicheli et al., 2020).

Circulating tumor cells (CTCs) refer to tumor cells detaching from the primary tumor or metastatic lesions and then entering into bloodstream (Fabisiewicz et al., 2020; Massagué & Obenauf, 2016). CTCs represent the undergoing process of metastases (Massagué & Obenauf, 2016). Therefore, in the updated TNM (tumor-node-metastasis) staging for BC, the National Comprehensive Cancer Network (NCCN) guidelines have added a new M0 (i+) category that is defined as “no clinical or radiographic evidence of distant metastases, but existence of detected tumor cells in the circulation fluids” (Gradishar et al., 2018). Clinical application of CTCs has been extensively exploited in BC. CTCs are good prognostic markers to predict patients survival outcome, and thus can be used to risk stratify patients and guide therapy (Fabisiewicz et al., 2020; Cristofanilli et al., 2005; Liu et al., 2009; Hwang, Hwang & Miyamoto, 2016). CTCs can be used as a diagnostic tool for screening and early diagnosis of BC and as a surveillance tool for real-time monitoring of disease progression and patients response to treatment (Jin et al., 2020; Yan et al., 2017). Many technologies have been developed to detect CTCs. However, for lacking a validated standard of CTC results from different platforms, CTC is not yet considered as a standard routine diagnostic tool used in clinical practice. Different CTC detection methodologies have different sensitivities, leading to varying CTC cut-off values for the same clinical application (Krawczyk et al., 2013). CellSearch® is so far the only CTC system approved by Food and Drug Administration (FDA). But CellSearch® had in general low CTC detection rates in BC, around 40–50% in metastatic BC and less than 30% in early BC (Eroglu, Fielder & SomLo, 2013). Based on the CellSearch® data on BC, the 8th edition of American Joint Committee on Cancer (AJCC) Cancer Staging Manual for BC has recognized CTC as a prognostic factor (Amin et al., 2017). Early BC patients with ≧1 CTC per 7.5 mL of blood or metastatic BC patients with ≧5 CTC per 7.5 mL of blood are considered as patients with higher risks for poor prognosis (Amin et al., 2017). With the improvement of new technologies, CTC detection has become more sensitive. As there is currently no prevailing CTC detection system, a standard must be established to compare results from different CTC detection platforms. With the ease of sample manipulation, the microfluidic-based strategies for CTC enrichment usually have a better sensitivity and specificity for CTC detection (Jin et al., 2020; Stott et al., 2010). CytoSorter® (Hangzhou Watson Biotech, Hangzhou, China), a microfluidic-based immuno-capture CTC platform, has been used in this study for CTC detection. CytoSorter® technology employs the positive selection utilizing a streptavidin nanoarray chip. The microfluidic chip can be coated with any biotin-labeled capture antibody to enrich desired cells. The most common CTC capture antibody is anti-epithelial cell adhesion molecules (EpCAM) CytoSorter® CTC system has been validated in BC (Jin et al., 2020), head and neck cancer (Zheng et al., 2019), pancreatic cancer (Wei et al., 2019) and lung cancer (Xie et al., 2021). CytoSorter® has CTC detection rates over 90% in stage II–III BC patients (Jin et al., 2020), much higher than the CellSearch® system (Eroglu, Fielder & SomLo, 2013). With a CTC cut-off of two per four mL of blood, CytoSorter® has a sensitivity and specificity of 76.56 % and 95.4%, respectively, for diagnosing BC (Jin et al., 2020).

It is reported that a single positive CTC result with CellSearch® 5 years later after curative treatment could be still considered as an independent prognostic factor for late recurrence in ER-positive BC (Sparano et al., 2018). In this retrospective study, we aimed at evaluating whether preoperative (pre-op) CTCs could be used as a prognostic biomarker for early recurrence of disease in BC, and at determining the CytoSorter® CTC cut-off value for the identification of high-risk BC patients for early recurrence.

Materials and Methods

Study design

Studies have shown that CTC enumerations prior to therapy, after initial treatment and during follow-ups could be all considered as a prognostic factor in metastatic BC (Fabisiewicz et al., 2020; Cristofanilli et al., 2005; Liu et al., 2009; Hwang, Hwang & Miyamoto, 2016). This study aimed to determine whether pre-op CTC could be used as a biomarker to predict early recurrence of disease in BC patients with resectable tumors. The first peak of early recurrence usually occurs in around 9 months after curative surgery and might be delayed to 18 months in patients with adjuvant therapies. Therefore, stage II-III BC patients who performed pre-op CTC detection prior to any treatment, and had a curative mastectomy or lumpectomy between January and May 2018 and follow-up records until December 31st 2019 were retrospectively enrolled in this study as shown in Fig. 1. Relapses during follow-up were recorded. Correlations of relapse and survival with CTCs, patients clinicopathological characteristics, and received adjuvant therapies would be analyzed. BC patients inclusion criteria were as follows: (1) female patients aged between 18 to 75 years; (2) patients who had CTC detection prior to surgery; (3) stage II or III BC patients with a resectable tumor; (4) patients who had surgery as the primary treatment; (5) patients who had negative history of other malignancy within five years prior to CTC detection; (6) patients were treatment-naive before CTC detection. Patients of the following descriptions were excluded and rejected from the study: (1) patients who were breast-feeding or pregnant at time of CTC detection or surgery; (2) patients who lost contact during follow-up; (3) patients with ductal carcinoma in situ (DCIS), stage I tumor, or metastatic cancer; (4) patients who had other conditions which were considered not suitable for the study.

Figure 1 The study design of this retrospective study.

The follow up ended on the 31st of December 2019. (pre-op = pre-operation; CTC = circulating tumor cells).

Patients and ethics

In total, 36 qualified BC patients were included for analysis. All patients were treated in accordance with the Chinese Society of Clinical Oncology (CSCO) guidelines for breast cancer (Huang & Yin, 2018). According to these guidelines, hormonal therapy was provided for patients with hormone receptor-positive diseases, while Trastuzumab treatment was indicated in HER2 positive patients except for those with a tumor smaller than 0.5 cm. Chemotherapy was indicated in node-positive patients, node-negative patients with a tumor larger than 2 cm, hormonal receptor-negative, or HER2 positive, and those with a grade 3 tumor. Table 1 displays the clinicopathological characteristics of the included patients. Among the 36 patients, 29 were at stage II while 7 were at stage III. Their ages were in the range between 34 and 70 years (mean and median of 53). A total of 75% of the patients were ER-positive, while 58.33% were PR-positive. 92.59% of the patients were hormone receptor-positive and thus received adjuvant hormonal therapy. All of 10 HER2 positive-patients received adjuvant treatment with trastuzumab. Twenty-nine and 23 out of the included 36 patients had adjuvant chemotherapy and radiation therapy, respectively. There was no difference in treatment between patients with different amount of CTCs since CTCs were not yet written in the guidelines to guide treatment. Follow-up was carried out in every 3 months in the first 2 years with an annual mammogram, which is also in accordance with the aforementioned guidelines. The principles established in the Declaration of Helsinki were followed in this study, which was approved by the ethics committee of Zhejiang University Medical College Affiliated Sir Run Run Shaw Hospital with IRB number, Qi Xie Lin Chuang Shi Yan 20180427-1. Written consent for the participation in the research and the publication of their case details was obtained from each patient.

Table 1 Correlation of recurrence and recurrence-free-survival (RFS) with patient demographics and clinicopathological characteristics.

Parameter	n	Recurrence	P	RFS (P)*	D-recurrence	P	D-RFS (P)*	
Age								
<53	19	7	0.1279	0.062	5	0.408	0.193	
≧53	17	2	2	
TNM stage								
II	29	5	0.0497	0.033	4	0.1159	0.093	
III	7	4	3	
T stage								
T1	5	1	0.2193	0.11	1	0.8124	0.666	
T2	28	6	5	
T3	3	2	1	
N stage								
N0	15	2	0.0643	0.083	2	0.0309	0.053	
N1	14	4	2	
N2	5	1	1	
N3	2	2	2	
Differentiation						
I	13	5	0.216	0.195	4	0.364	0.327	
II	11	3	2	
III	12	1	1	
ER						
Yes	27	7	1	0.855	5	1	0.818	
No	9	2	2	
PR								
Yes	21	5	1	0.919	5	0.6738	0.452	
No	15	4	2	
HER2						
Yes	10	4	0.2262	0.134	3	0.3696	0.237	
No	26	5	4	
Menopause							
Yes	21	5	1	0.754	3	0.4178	0.316	
No	15	4	4	
Ad chemo TX							
Yes	29	8	0.6518	0.374	7	0.3029	0.13	
No	7	1	0	
Ad rad TX							
Yes	23	8	0.1136	0.109	7	0.0343	0.047	
No	13	1	0	
Ad hor TX							
Yes	28	7	1	0.945	5	0.639	0.637	
No	8	2	2	
Ad tar TX							
Yes	10	4	0.2262	0.134	2	1	0.134	
No	26	5	5	
CTCs							
≧4	9	5	0.0262	0.013	4	0.0497	0.029	
<4	27	4	3	
Notes:

* Follow-up (days): min = 128; max = 723; mean = 518; median = 536.

Abbreviation: n = number; RFS = recurrence-free survival; D = distant; D-RFS = distant recurrence-free survival; TNM = tumor-node-metastasis; ER = estrogen receptor; PR = progesterone receptor; HER2 = human epidermal growth factor receptor-2; Ad chemo TX = adjuvant chemotherapy; Ad rad TX = adjuvant radiation therapy; Ad hor TX = adjuvant hormonal therapy; Ad tar TX = adjuvant targeted therapy; CTCs = circulating tumor cells.

CTC detection

CytoSorter® EpCAM epithelial CTC kit was used for CTC detection. Four mL of peripheral blood was collected for CTC detection before surgery. The CTC detection procedure followed the manufacturer protocol and was described as in the previous study (Jin et al., 2020). In brief, peripheral blood mononuclear cells (PBMCs) were acquired first via Histopaque®-1077 (Sigma–Aldrich, Shanghai, China) density gradient centrifugation, and then the PBMC fraction was washed and loaded into CytoSorter® system. Potential CTCs were captured on an EpCAM antibody immobilized microfluidic chip. The microfluidic chip was taken away from CytoSorter® once the CTC enrichment was finished. This was followed by the immunofluorescence staining of PanCK-FITC (pancytokeratin-fluorescein isothiocyanate), CD45-PE (cluster of differentiation 45-phycoerythrin) and DAPI (4,6-diamidino-2-phenylindole). An OPPNO immunofluorescence microscope (DSY5000X; OPPNO, Chongqing, China) was used to identify CTCs by searching for CD45-PE−, PanCK-FITC+, and DAPI+ cells. The imaging results were double-checked independently by two experienced technicians.

Statistical analysis

Statistical analyses were performed using Prism 6.0 (Graphpad, La Jolla, CA, USA) and SPSS 20 (IBM Corp., Armonk, NY, USA). A data-set containing patients clinicopathological characteristics, demographics, CTC results, adjuvant therapy regimens as well as surgery and recurrence dates was created. The following patient clinicopathological characteristics were included: age, TNM stage, statuses of ER, progesterone receptor (PR) and HER2, differential grade of the tumor based on the Bloom-Richardson method, menopausal status and location of recurrence (local vs distant) if occurred. Recurrence free survival (RFS) time was defined as the duration between the date of surgery and the date of objectifying the recurrence with a suitable diagnostic test, while distant RFS (D-RFS) was defined as the time between the date of surgery and the date on which the distant recurrence was confirmed. Follow-up time was defined as the duration between the the date of surgery and the last checkup. Patients considered as free of recurrence were those who showed no objectified recurrence at the end of follow-up. The x2 test and Fisher’s exact test were used for comparing the categorical parameters. The cut-off value of CTCs for relapse was determined by statistical methods and Youden index. The cut-off value should be able to divide patients into two groups with statistically significant differences of relapses and have the highest Youden index. Factors influencing the RFS and D-RFS were identified first by the univariate Kaplan-Meier log-rank test, followed by assessments in the multivariate logistic Cox regression model to identify independent predictors of both RFS and D-RFS. Kaplan-Meier curves were generated for RFS and D-RFS. A two-sided p value of less than 0.05 was considered to be statistically significant.

Results

Correlation of patients characteristics with recurrence and distant recurrence

CTCs were detected in 32 out of 36 enrolled patients. CTCs were associated with cancer stage and tumor size as shown in Table S1 (P = 0.0462 and 0.027, respectively). Nine patients developed relapses during follow-up, and 7 of them were distant recurrence. Common distant recurrence sites were bones, liver, lung and brain. None of the included patients died during follow-up. Correlation of patient characteristics and CTCs with relapses is shown in Table 1. Relapse was associated with cancer stage (P = 0.0497) and CTCs (P = 0.0224). Patients with relapses tend to have a higher pre-op CTCs of 3.78 (median: 4) per four mL of blood than those without (2.15, median: 2). Relapse was not correlated with tumor differentiation state and molecular subtype, nor received adjuvant therapies (all P > 0.05). Distant recurrence was associated only with lymph node involvement (P = 0.0309). Only when CTC cut-off for relapse was set to four, patients could be divided into two groups with statistical significance as shown in Table S2. If patients with CTCs ≧4 was defined as CTC positive for relapse, CTC positive rates were correlated with relapse and distant recurrence (P = 0.0262 and 0.0497, respectively).

Univariate analysis for RFS and D-RFS

The significance of patient clinicopathological characteristics with RFS and D-RFS are shown in Table 1. Only cancer stage and CTCs were significant for RFS (P = 0.033 and 0.013, respectively). Time to relapse ranged between 128 and 566 days. Six patients showed recurrence within 1 year after surgery. Mean (median) of follow-up was 518 (536) days. Patients with ≧4 CTCs per four mL of blood showed a significantly reduced RFS compared to patients with <4 CTCs (log-rank P = 0.013). As for D-RFS, only adjuvant radiation therapy and CTCs showed significance (log-rank P = 0.047 and 0.029, respectively). The 2-year RFS and D-RFS Kaplan-Meier survival curves are shown in Fig. 2. Patients with <4 CTCs had RFS and D-RFS of 85.2% and 88.9%, respectively, while patients with ≧4 CTCs had reduced RFS and D-RFS of 44.4% and 55.6%, respectively. When CTC cut-off was set to four, CTC had a sensitivity and specificity of 55.6% and 85.2%, respectively, for relapse. Patients with stage III tumors had a reduced RFS of 42.9% compared to stage II patients (82.8%, log-rank P = 0.033).

Figure 2 Kaplan–Meier plots for patient outcomes.

Difference in recurrence-free survival (RFS) (A) and distant recurrence-free survival (D-RFS) (B) were compared in risk groups stratified by CTC cut-off value of four per four mL of blood. Patients with CTCs ≧ 4 have reduced RFS and D-RFS compared to patients with CTCs < 4 (P = 0.013 and 0.029, respectively). (CTC = Circulating tumor cell).

Multivariate analysis for RFS and D-RFS

Cox regression analysis for RFS and D-RFS was performed. All significant univariate factors (P < 0.1) were included to identify the true independent predictors of relapse and distant recurrence. After excluding the non-significant variables in a stepwise manner, the final model is shown in Table 2. CTC with a cut-off value of four is the only independent prognostic factor identified for both RFS and D-RFS (P = 0.024 and 0.047, respectively).

Table 2 Final model of multivariate Cox regression analysis for prediction of recurrence-free survival or distant recurrence-free survival among univariately significant parameters.

Parameter	HR	95% CI	P	
RFS				
CTCs < 4 vs ≧ 4	0.219	[0.058–0.82]	0.024	
D-RFS				
CTCs < 4 vs ≧ 4	0.218	[0.048–0.977]	0.047	
Note:

Abbreviation: HR = hazard ratio; CI = confidence interval; RFS = recurrence-free survival; CTCs = circulating tumor cells; D-RFS = distant recurrence-free survival.

Discussions

Studies have shown that CTCs could be considered as an independent prognostic factor for progression-free and overall survival (PFS and OS) in metastatic BC (Cristofanilli et al., 2005; Liu et al., 2009). In the 8th edition of AJCC Cancer Staging Manual for BC, it is written that with ≧1 and ≧5 CTCs per 7.5 mL of peripheral blood indicate poor prognosis in early and metastatic BC, respectively (Amin et al., 2017). Furthermore, a study concerning pre-op CTC enumeration and prognosis showed as well that an increased risk for distant recurrence and BC-related death is associated with the presence of pre-op CTCs in BC patients (Franken et al., 2012). Most studies investigating CTCs and recurrence in BC had follow-up time usually longer than 5 years (Cristofanilli et al., 2005; Liu et al., 2009; Stott et al., 2010; Franken et al., 2012). They were more looking at the overall relapses and late recurrence. Results of these studies indicated that CTCs could be used as a prognostic factor for RFS, D-RFS and late recurrence (Cristofanilli et al., 2005; Liu et al., 2009; Stott et al., 2010; Franken et al., 2012). The first peak of early recurrence of BC occurs within 2 years after initial treatment (Colleoni et al., 2016; Demicheli et al., 2020). The aim of this study was to evaluate the use of pre-op CTCs as prognostic marker to identify high-risk BC patients for early recurrence. Our results show that CTC with cut-off value of four with CytoSorter® system is an independent prognostic factor for 2-year RFS and D-RFS, suggesting that CTCs can be used as a predictor for early recurrence of disease in BC.

A CTC cut-off value of four was found in our study for the identification of high-risk patients for early recurrence. As different CTC capturing systems have varying sensitivities, different CTC cut-off values for recurrence were reported. Franken et al used CellSearch® and found a CTC cut-off value of 1 per 30 mL of blood can be used as a predictor for recurrence, distant recurrence and BC-related death (Franken et al., 2012). Sparano et al found that using CellSearch®, CTCs with a cut-off value of 1 per 7.5 mL could be used to predict the late recurrence in ER-positive BC patients (Sparano et al., 2018). CTC cut-off values depends on patients status, clinical applications and CTC detection methods. The cut-off value would be higher in patients at advanced tumor stage, as AJCC staging manual uses CTC cut-off values of 1 and 5 per 7.5 mL of blood, respectively, in early and metastatic BC for prognosis (Stott et al., 2010). Same system may use different CTC cut-off values for different clinical applications. The previous study with CytoSorter® system showed that CTCs with a cut-off value of two per four mL of blood could be used as a diagnostic aid for diagnosing BC with a sensitivity and specificity of 76.56% and 95.4%, respectively (Jin et al., 2020). However, results of this study show that the cut-off value of pre-op CTCs for early recurrence in BC with CytoSorter® system should be four CTCs per four mL of blood. Different CTC systems use different strategies to detect CTCs, thus resulting in different sensitivities regarding CTC detection rates. It is reported that CellSearch® has CTC detection rates less than 40%, while CytoSorter® has CTC detection rates more than 80% in non-metastatic BC (Eroglu, Fielder & SomLo, 2013; Jin et al., 2020). Although CellSearch® and CytoSorter® EpCAM epithelial CTCs kit are both meant to capture the EpCAM-positive epithelial CTCs, they should have different CTC cut-off values for predicting recurrence. As CytoSorter® is more sensitive than CellSearch®, it should have a higher CTC cut-off value for poor prognosis. Patients enrolled in this study were at either stage II or stage III, in between early and metastatic BC. According to the AJCC staging manual, the CTC cut-off value for poor prognosis in enrolled patients should be in between one and five per 7.5 mL of blood. And we found a CTC cut-off value of four per four mL of blood for early recurrence. Four CTCs per four mL of blood equals to 7.5 CTCs per 7.5 mL of blood, which is in line with our hypothesis that CytoSorter® should have a higher CTC cut-off value for poor prognosis than CellSearch®.

Our results showed that only CTC was associated with 2-year RFS and D-RFS, while patient demographics and clinical characteristics, and adjuvant therapies were not. recurrence were surprisingly more common in patients receiving adjuvant therapy. It may be due to the small sample size of the included patients, or adjuvant therapy might be more effective in reducing late recurrence rather than early recurrence. Also, patients requiring adjuvant therapy are usually patients at higher risks for disease progression. Therefore, they might more easily develop early recurrence.

Some BC patients are considered as high-risk patients for they develop recurrence easily after conventional treatment. To improve the curative rate of BC, a precise stratification of patients that will benefit from different customized therapies is required. Recurrence of disease may originate from the minimal residue disease of a primary tumor, or from another hidden lesion site which was not discovered at diagnosis (Klein, 2009). Micrometastasis in the bone marrow is a high-risk factor for disease recurrence, and can be found in about 30% of BC patients (Braun et al., 2000; Braun et al., 2005). However, screening for occult metastatic tumor cells in the bone marrow is not yet considered as a standard procedure in clinical routine due to being too invasive (Fehm et al., 2006). Routinely used biomarkers for discriminating between low- and high-risk patients include serum tumor makers array and histological staining of biopsy samples for ER, PR, HER2 and Ki67 (Amin et al., 2017). The results are sometimes not so satisfying, for distant metastasis may cause deaths in some low-risk patients, whereas some high-risk patients would survive for decades (Fabisiewicz et al., 2020; Cristofanilli et al., 2005). An accurate method to stratify patients with different risks is still in need. Detecting tumor cells in the blood is a promising alternative for it is non-invasive and can be performed frequently. It is reported that tumor cells can be released into bloodstream even when the tumor lesion is smaller than a sesame and hardly detected by the routine used medical imaging techniques (Hu et al., 2019). Therefore, CTCs can be used as a dynamic monitoring tool to tell the presence of recurrence even before imaging methods can detect. Our results indicate that pre-op CTC detection can be used as well to predict early recurrence of disease, therefore enabling physicians to identify high-risk patients and provide a different enhanced treatment to increase their curative rate. When CTC cut-off was set to four, CTC had a sensitivity and specificity of 55.6% and 85.2%, respectively, for early recurrence. However, the sensitivity and specificity are not quite satisfying. Pre-op CTC detection should be combined with other prognostic evaluations, such as ER, PR, HER2 and Ki67, to better stratify patients with different risks.

Taken together, CTCs can be used as a prognostic factor and monitoring tool for recurrence. The major drawbacks of this study are that this is a retrospective study with limited sample size, too heterogeneous sample and short follow-up. Only 36 BC patients with diverse background were included for analysis and the median follow-up was 518 days. Patients heterogeneity led to small sample sizes in each sub-group and thus resulted in some misleading results without statistical significance. A longer follow-up may have increased the percentage of recurrence. Furthermore, most patients had only performed the CTC detection once before surgery. Since it is believed that minimal residual disease (MRD) is the key to recurrence (Lin, Orozco & Hoon, 2018), it would be more interesting to look for the minimal residual disease by postoperative CTC detection.

Conclusions

Results of this study show pre-op CTC detection could be used as an independent prognostic factor for 2-year RFS and D-RFS. A CytoSorter® CTC cut-off value of four per four mL of blood is found to identify high-risk BC patients for early recurrence. Still a prospective study with a larger patient population with more homogeneous background and longer follow-up is needed to verify our findings.

Supplemental Information

Supplemental Information 1 Raw data.

Deidentified patient demographics and clinical characteristics and follow-up information

Click here for additional data file.

Supplemental Information 2 Correlation of CTCs with patient demographics and clinicopathological characteristics.

Click here for additional data file.

Supplemental Information 3 Correlation of recurrence and recurrence-free-survival (RFS) with CTCs with different cut-off values.

Click here for additional data file.

Additional Information and Declarations

Competing Interests

Author Contributions

Human Ethics

Data Availability

Wan-Hung Fan and Meiqiong Wu are currently employees of Hangzhou Watson Biotech. The authors declare that they have no competing interests.

Lidan Jin conceived and designed the experiments, performed the experiments, analyzed the data, prepared figures and/or tables, authored or reviewed drafts of the paper, and approved the final draft.

Wan-Hung Fan conceived and designed the experiments, analyzed the data, prepared figures and/or tables, authored or reviewed drafts of the paper, and approved the final draft.

Yi Luan performed the experiments, prepared figures and/or tables, and approved the final draft.

Meiqiong Wu performed the experiments, prepared figures and/or tables, and approved the final draft.

Wenhe Zhao conceived and designed the experiments, authored or reviewed drafts of the paper, and approved the final draft.

The following information was supplied relating to ethical approvals (i.e., approving body and any reference numbers):

The ethics committee of Zhejiang University Medical College Affiliated Sir Run Run Shaw Hospital (approval number Qi Xie Lin Chuang Shi Yan 20180427-1).

The following information was supplied regarding data availability:

Data are available in the Supplemental File.

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
