# Peer review of "Evaluation of circulating tumor cells as a prognostic biomarker for early recurrence in stage II–III breast cancer patients using CytoSorter® system: a retrospective study"

_PeerJ, doi:10.7717/peerj.11366_

## Round 0.1 · original submission · Major Revisions

You will see from the comments of the two expert reviewers that they have identified a number of serious issues that you will need to take into account, ranging from the validity of the cut-off point, the limited number of samples, the multi-variate analysis and the comparison with the currently approved methodology.

These are all helpful and constructive criticisms and please address all of these points (and all of the other issues raised) if you wish to revise your manuscript.

Reviewer 1 ·

Basic reporting

no comment

Experimental design

The method used in the study is remarkably efficient comparing to the other systems, but there is a very little information about it in the study itself, but also generally in the literature. It should be better described (and probably deserves some comparative study with, for example, CellSearch system?).

Validity of the findings

1. The group of 36 patients in the study is very heterogeneous in almost every parameter. This is not optimal for the analysis of such a small group of patients. Some correlations may not manifest themselves, because the numbers in the analyzed subgroups are too small to find significance. It makes the study less reliable.
The description of the multivariate analysis is somehow cryptic and it is not clear how many parameters were included (“all univariate significant factors”, which means two? That’s hardly multivariate). On the other hand, fourteen is too many. This needs clarification.

2. It is not clear how the authors came up with the cutoff value of 4? Was the receiver operating characteristics performed?

Additional comments

In the manuscript entitled „Evaluation of circulating tumor cells as a prognostic biomarker for early recurrence in stage II-III breast cancer patients using CytoSorter® system” the authors address a very important issue of finding new ways to stratify patients with early breast cancer to assess the probability of metastasis, which is crucial for choosing the treatment options. The authors use a very efficient new method and their results imply that the pre-operative CTC count may be useful in the identification of BC patients with a higher risk for early recurrence. While the whole manuscript is well written and compelling, there are some issues that need to be addressed (details in sections: experimental design, validity of the findings).

Minor comments:
1. The caption of the Figure 1 is slightly incorrect, since the figure presents not only Kaplan-Meier curves but also (representative?) images of the identified CTCs.
2. The description of patients clinicopathological characteristics is in Table 1 not Table 2 (verse 172).
3. The citation of Bidard et al. (ref 17) to show that CytoSorter has a higher CTC detection rate than CellSearch is not correct since it does not mention CytoSorter and even general statements about microfluidics are not that categorical.

Reviewer 2 ·

Basic reporting

English used throughout the paper is sometimes incorrect, as well as there are some sentences difficult to understand and spelling mistakes.ex: lines 54, 75-77,82,116...

Some concepts should need more details, such as "triple negative" to make it comprehensive to the readers.

In some sentences references are needed. ex: line 68, 236, 238

The statement in line 78-79 should be revised.

Experimental design

Some of the parameters for the selected patients included in the study present bias: T and N stage, Hormone status, different treatments and CTCs. It would be better to compensate the groups.

The number of patients incuded in the study is very small. The cut-off value (4) should be validated in a larger number of patients.

Validity of the findings

In the results section "correlation of patients characteristics with recurrence and distance recurrence", some information regarding the clinicopathological characteristics is detailed (line 159-165). This information would fit better in the materials and methods section, in order to not confuse the readers with too much number information.

The cut-off value set up has a specificity of 55,6%, which means that almost half of the patients would be diagnosed as a positive being negative for the recurrence. This misclassification could generate over treatment of the patients among others.

In the conclusions the authors don't mention the disadvantages of using an EpCAM enriching device. Moreover, since the technology used to detect CTCs is quite similar to the already method aproved by the FDA (CellSeacrch), it would be interesting to make a point (less confusing than lines 217-221) of the potential advantages of CytoSorter over CellSearch.

---

## Round 0.2 · accepted · Accept

Many thanks for revising the manuscript according to the reviewers' recommendations. They are content with those changes.

Reviewer 1 ·

Basic reporting

I have no further comments.

Experimental design

My comments have been addressed. I have no further comments.

Validity of the findings

My comments have been addressed. I have no further comments.

Additional comments

I have no further comments.

Reviewer 2 ·

Basic reporting

no comment

Experimental design

no comment

Validity of the findings

no comment

Additional comments

Dear Authors,
Thanks for the clarification and for introducing changes in the manuscript. I would recommend the new version of the manuscript to be published. However due to the impact that CTC can have in clinics it would be interesting to validate these results in a large number of patients, as well as in a more homogeneous samples.
Sincerely,